# Coordinated Multi-Agent Exploration using Shared Goals

## Abstract

Exploration is critical for good results of deep reinforcement learning algorithms and has attracted much attention. However, existing multi-agent deep reinforcement learning algorithms still use mostly noise-based techniques. It was recognized recently that noise-based exploration is suboptimal in multi-agent settings, and exploration methods that consider agents' cooperation have been developed. However, existing methods suffer from a common challenge: agents struggle to identify states that are worth exploring, and don't coordinate their exploration efforts toward those states. To address this shortcoming, in this paper, we proposed coordinated multi-agent exploration (CMAE): agents share a common goal while exploring. The goal is selected by a normalized entropy-based technique from multiple projected state spaces. Then, agents are trained to reach the goal in a coordinated manner. We demonstrated that our approach needs only $1\% - 5\%$ of the environment steps to achieve similar or better returns than state-of-the-art baselines on various sparse-reward tasks, including a sparse-reward version of the Starcraft multi-agent challenge (SMAC).

## 1 Introduction

Cooperative multi-agent reinforcement learning (MARL) is an increasingly important field. Indeed, many real-world problems are naturally modeled using MARL techniques. For instance, tasks from areas as diverse as robot fleet coordination (Swamy et al., 2020; Hüttenrauch et al., 2019) and autonomous traffic control (Bazzan, 2008; Sunehag et al., 2018) fit MARL formulations.

To address MARL problems, early work followed the independent single-agent reinforcement learning paradigm (Tampuu et al., 2015; Tan, 1993; Matignon et al., 2012). However, more recently, specifically tailored techniques such as monotonic value function factorization (QMIX) (Rashid et al., 2018), multi-agent deep deterministic policy gradient (MADDPG) (Lowe et al., 2017), and counterfactual multi-agent policy gradients (COMA) (Foerster et al., 2018) have been developed. Those methods excel in a multi-agent setting because they address the non-stationary issue of MARL and develop communication protocols between agents. Despite those advances and the resulting reported performance improvements, a common issue remained: all of the aforementioned methods use exploration techniques from classical algorithms. Specifically, these methods employ noise-based exploration, *i.e.*, the exploration policy is a noisy version of the actor policy. For instance, Lowe et al. (2017) add Ornstein-Uhlenbeck (OU) (Uhlenbeck & Ornstein, 1930) noise or Gaussian noise to the actor policy. Foerster et al. (2016); Rashid et al. (2018); Yang et al. (2018); Foerster et al. (2017) use variants of $\epsilon$-greedy exploration, where a random suboptimal action is selected with probability $\epsilon$.

It was recognized recently that use of classical exploration techniques is sub-optimal in a multi-agent reinforcement learning setting. Specifically, Mahajan et al. (2019) show that QMIX with $\epsilon$-greedy exploration results in slow exploration and sub-optimality. Mahajan et al. (2019) improve exploration by conditioning an agent's behavior on a shared latent variable controlled by a hierarchical policy. Even more recently, Wang et al. (2020) encourage coordinated exploration by considering the influence of one agent's behavior on other agents' behaviors.

While all of the aforementioned exploration techniques for multi-agent reinforcement learning significantly improve results, they suffer from a common challenge: agents struggle to identify states that are worth exploring, and don't coordinate their exploration efforts toward those states. To give

an example, consider a push-box task, where two agents need to jointly push a heavy box to a specific location before observing a reward. In this situation, instead of exploring the environment independently, agents need to coordinate pushing the box within the environment to find the specific location.

To address this issue, we propose coordinated multi-agent exploration (CMAE), where multiple agents share a common goal. We achieve this by first projecting the joint state space to multiple subspaces. We develop a normalized entropy (Cover & Thomas., 1991)-based technique to select a goal from the under-explored subspaces. Then, exploration policies are trained to reach the goals in a coordinated manner.

To show that CMAE improves results, we evaluate our approach on various environments with sparse-rewards from Wang et al. (2020), and the sparse-reward version of the Starcraft multi-agent challenge (SMAC) (Samvelyan et al., 2019), which requires coordinated actions among agents over extended time steps before observing a reward. The experimental results show that our approach needs only $1\% - 5\%$ of environment steps to achieve similar or better average test episode returns than current state-of-the-art baselines.

## 2 PRELIMINARIES

In this section, we define the multi-agent Markov decision process (MDP) in Sec. 2.1, and introduce the multi-agent reinforcement learning setting in Sec. 2.2.

### 2.1 MULTI-AGENT MARKOV DECISION PROCESS

We model a cooperative multi-agent system as a multi-agent Markov decision process (MDP). An $n$-agent MDP is defined by a tuple $G = (\mathcal{S}, \mathcal{A}, \mathcal{T}, \mathcal{R}, \mathcal{Z}, \mathcal{O}, n, \gamma, H)$. $\mathcal{S}$ is the global state space of the environment. $\mathcal{A}$ is the action space. At each time step $t$, each agent's policy $\pi_i$, $i \in \{1, \dots, n\}$, selects an action $a_i^t \in \mathcal{A}$. All selected actions form a joint action $\boldsymbol{a}^t \in \mathcal{A}^n$. The transition function $\mathcal{T}$ maps the current state $s^t$ and the joint action $\boldsymbol{a}^t$ to the next state $s^{t+1}$, *i.e.*, $\mathcal{T} : \mathcal{S} \times \mathcal{A}^n \to \mathcal{S}$. All agents receive a collective reward $r^t \in \mathbb{R}$ according to the reward function $\mathcal{R} : \mathcal{S} \times \mathcal{A}^n \to \mathbb{R}$. The goal of all agents' policies is to maximize the collective expected return $\sum_{t=0}^{H} \gamma^t r^t$, where $\gamma \in [0, 1]$ is the discount factor, $H$ is the horizon, and $r^t$ is the collective reward obtained at timestep $t$. Each agent $i$ observes local observation $o_i^t \in \mathcal{Z}$ according to the observation function $\mathcal{O} : \mathcal{S} \to \mathcal{Z}$. Note, observations usually reveal partial information about the global state. For instance, suppose the global state contains the location of agents, while the local observation of an agent may only contain the location of other agents within a limited distance. All agents' local observations form a joint observation, denoted by $\boldsymbol{o}^t$.

A global state space $\mathcal{S}$ is the product of component spaces $V_i$, *i.e.*, $\mathcal{S} = \prod_{i=1}^{M} V_i$, where $V_i \subseteq \mathbb{R}$ (Samvelyan et al., 2019; Lowe et al., 2017; Rashid et al., 2018; Foerster et al., 2018; Mahajan et al., 2019). We refer to $V_i$ as a 'state component.' The set of all component spaces of a product space is referred to as the component set. For instance, the component set of $\mathcal{S}$ is $\{V_i | i \in \{1, \dots, M\}\}$. Each entity, *e.g.*, agents, objects, *etc.*, in the environment are described by a set of state components. We refer to a set of state components that is associated with an entity in the environment as an 'entity set.' For instance, in a 2-agent push-box environment, where two agents can only collaboratively push a box to a goal location, we have the global state space $\mathcal{S} = \prod_{i=1}^{6} V_i$, where $\{V_1, V_2\}, \{V_3, V_4\}, \{V_5, V_6\}$ represent the location of agent one, agent two, and the box, separately. Consequently, $\{V_1, V_2\}, \{V_3, V_4\}, \{V_5, V_6\}$ are three entity sets.

### 2.2 MULTI-AGENT REINFORCEMENT LEARNING

In this paper, we follow the standard centralized training and decentralized execution (CTDE) paradigm (Lowe et al., 2017; Rashid et al., 2018; Foerster et al., 2018; Mahajan et al., 2019). That is, at training time, the learning algorithm has access to all agents' local observations, actions, and the global state. At execution time, *i.e.*, at test time, each individual agent's policy only has access to its own local observation.

The proposed CMAE is applicable to off-policy MARL methods (*e.g.*, Rashid et al., 2018; Lowe et al., 2017; Sunehag et al., 2018; Matignon et al., 2012). In off-policy MARL, exploration poli-

---

**Algorithm 1:** Training with Coordinated Multi-Agent Exploration (CMAE)

---

Initialize exploration policies $\{\mu_i\}_{i=1}^n$, target policies $\{\pi_i\}_{i=1}^n$, counters $\{c_k\}_{k=1}^K$;
Initialize the environment and replay buffer $\mathcal{D}$;
Initialize $\alpha = 1$;
**for** *episode* $= 1 \ldots M$ **do**
    Reset the environment, and observe global states $s^t$ and observations $\boldsymbol{o}^t = (o_1^t \ldots o_n^t)$;
    **for** $t = 1 \ldots T$ **do**
        UpdateCounter($\{c_k\}_{k=1}^K$, $s^t$, $\boldsymbol{o}^t$) ;
        Select actions $\boldsymbol{a}^t$ **using a mixture of exploration and target policies** $\alpha\mu_i + (1-\alpha)\pi_i$,
          $\alpha$ decreases linearly to 0;
        Apply $\boldsymbol{a}^t$ to the environment;
        Observe rewards $r^t$, state $s^{t+1}$, and local observations $\boldsymbol{o}^{t+1}$;
        Add transition tuple $\{s^t, \boldsymbol{o}^t, \boldsymbol{a}, s^{t+1}, \boldsymbol{o}^{t+1}, r^t\}$ to $\mathcal{D}$;
        TrainTarget($\{\pi_i\}_{i=1}^n$, $\mathcal{D}$);
    **end**
    TrainExp($\{\mu_i\}_{i=1}^n$, $\{c_k\}_{k=1}^K$, $\mathcal{D}$);
**end**

---

cies $\mu_i$, $i \in \{1, \ldots, n\}$ are responsible for collecting data from the environment. The data in the form of transition tuple $(s^t, \boldsymbol{o}^t, \boldsymbol{a}^t, s^{t+1}, \boldsymbol{o}^{t+1})$ is stored in a replay memory $\mathcal{D}$, *i.e.*, $\mathcal{D} = \{(s^t, \boldsymbol{o}^t, \boldsymbol{a}^t, s^{t+1}, \boldsymbol{o}^{t+1})\}_t$. The target policies are trained using transition tuples from the replay memory.

## 3    COORDINATED MULTI-AGENT EXPLORATION (CMAE)

In the following we first present an overview of CMAE before we discuss the method more formally.

**Overview:** The goal is to train the target policies $\{\pi_i\}_{i \in \{1, \ldots, n\}}$ of $n$ agents to maximize the environment episode return. Classical off-policy algorithms (Lowe et al., 2017; Rashid et al., 2018) typically use a noisy version of the target policies $\pi_i$ as the exploration policies $\mu_i$, *i.e.*, to collect data actions are selected based on exploration policies $\mu_i$. In contrast, in CMAE, we propose to train the exploration policies by training with a modified reward. Specifically, target polices are trained to maximize the usual external episode return. In contrast, exploration policies are trained to collect data from subspaces that haven't been well explored. We find this strategy to significantly improve training of target policies in the multi-agent reinforcement learning setting because this strategy can encourage multiple agents to jointly explore configurations of the state space.

Alg. 1 summarizes this approach. At each step, a mixture of the exploration policies $\{\mu_i\}_{i=1}^n$ and target policies $\{\pi_i\}_{i=1}^n$ are used to select actions. The resulting experience tuple is then stored in a replay memory $\mathcal{D}$. The target policies are trained directly using the data within the replay memory $\mathcal{D}$ at each step. Note that the exploration policies will only be updated at the end of each episode using a reshaped reward that encourages exploration polices to explore under-explored subspaces in a collaborative manner.

In the following we will provide details about how we propose to train the exploration policies.

### 3.1    TRAINING OF EXPLORATION POLICIES

To train the exploration policies $\mu_i$, $i \in \{1, \ldots, n\}$ we use a modified reward $\hat{r}$. This modified reward specifies the goal of the exploration. For example, in the two-agent push-box task, we specify a specific joint location of both agents and the box as a goal. Note, the agents will ignore all external rewards and only see positive reward when the goal, *i.e.*, the specified position is reached. The reward for the goal is set to $b$ while the rewards for all other situations are zero.

To find the goal situation we use $K$ counters $c_k$, $k \in \{1, \ldots, K\}$. A counter $c_k$ operates on a low-dimensional subspace $\mathcal{S}_k$ of the state space $\mathcal{S}$, *i.e.*, $\mathcal{S}_k \subseteq \mathcal{S}$. Occurrence of every configuration $s_k \in \mathcal{S}_k$ within the low-dimensional subspace will be recorded using the current replay buffer $\mathcal{D}$.

---

**Algorithm 2:** Train Exploration Policies (TrainExp)

---

Input: exploration policies $\{\mu_i\}_{i=1}^n$, counters $\{c_k\}_{k=1}^K$, replay buffer $\mathcal{D}$;
Initialize bonus $b$;
Compute normalized entropy $\eta(k)$ of subspace $k$ based on associated counter $c_k$;
$k^* = \arg\min_k \eta(k)$;
Sample a batch $B = \{s_i\}_{i=1}^M$ from $\mathcal{D}$;
$g = \arg\min_{s \in B} c_{k^*}(\text{proj}_{k^*}(s))$;
**for** $\{s^t, \boldsymbol{o}^t, \boldsymbol{a}, s^{t+1}, \boldsymbol{o}^{t+1}, r^t\} \in \mathcal{D}$ **do**
    **if** $s^t == g$ **then**
        $r^t = b$;
    **else**
        $r^t = 0$;
    **end**
    Update $\{\mu_i\}_{i=1}^n$ by $\{s^t, \boldsymbol{o}^t, \boldsymbol{a}, s^{t+1}, \boldsymbol{o}^{t+1}, r^t\}$
**end**

---

Let $\text{proj}_k$ be the projection from global state space to the subspace $k$. Formally, we obtain

$$c_k(s_k) = \sum_{s \in \mathcal{D}} \mathbb{1}[\text{proj}_k(s) = s_k],$$

where $\mathbb{1}[\cdot]$ is the indicator function (1 if argument is true; 0 otherwise) and $\text{proj}_k(s)$ denotes the restriction of state $s \in \mathcal{S}$ to subspace $\mathcal{S}_k$. Note that we are successively incrementing the counts, *i.e.*, the counters $c_k$ are not recomputed from scratch every time we train the exploration policies.

We subsequently normalize the counters $c_k(s_k)$ into a probability distribution $p_k(s_k) = c_k(s_k)/\sum_{\hat{s}_k \in \mathcal{S}_k} c_k(\hat{s}_k)$ which is then used to compute a normalized entropy $\eta_k = H/H_{\max} = -(\sum_{s \in \mathcal{S}_k} p_k(s) \log p_k(s))/\log(|\mathcal{S}_k|)$. We select the subspace $k^*$ with the smallest normalized entropy. From this subspace we choose the joint goal state $g$ by first sampling a batch of states $B$ from the replay buffer. From those states we select in a second step the state with the smallest count as the goal state $g$, *i.e.*, $g = \arg\min_{s \in B} c_{k^*}(s)$. Sampling of states is performed in order to avoid selecting unreachable states as a goal, *i.e.*, we encourage to explore states that we have seen rarely but at least once.

Given the goal state $g$, we train the exploration policies $\mu_i$ using the replay buffer $\mathcal{D}$ modified by a revised reward $\hat{r} = b$ if $s_{k^*} = g$. Note, $\hat{r} = 0$ otherwise. Consequently, the exploration policies $\mu_i$ focus exclusively on achieving the desired goal $g$. This strategy is summarized in Alg. 2.

As an alternative to the aforementioned subspace selection method, one could use probabilistic subspace selection, where the probability of a subspace being chosen is inversely proportional to its normalized entropy. The two different subspace selection approaches result in different exploration behaviors. Specifically, the probabilistic subspace selection will encourage exploration policies to explore more subspaces while the smallest normalized entropy method focuses on the most underexplored subspace.

## 3.2 SELECTING SUBSPACES

Which $K$ subspaces do we choose? As discussed in Sec. 2.1, we assume the global state space $\mathcal{S}$ to be composed out of a set of $M$ component spaces $V_i$. The number of possible subspaces is equivalent to the size of the powerset, *i.e.*, $2^M$. This is clearly intractable.

To address this, we select a subset of subspaces in levels. In each level $l$, we consider $l$ entities jointly. Recall, that entities are agents, objects, *etc.*, that are represented within the state space $\mathcal{S}$. Suppose the global state space has $N$ entity sets $A_1, \ldots, A_N$. In level $l \leq N$, a subspace's component set is the union of $l$ distinct entity sets. Formally, let $D_E$ be a component set of a subspace in level $l$, we have

$$D_E = \bigcup_{i \in E} A_i, \forall E \in \binom{\{1, \ldots, N\}}{l},$$

where $\binom{\{1,\cdots,N\}}{l}$ represents the set of all $l$-combinations of $\{1, \ldots, N\}$.

Note that there are many equivalent component sets in a level, if agents are homogeneous, *i.e.*, if agents have identical action and observation space and are controlled by the same policy.

To see this, consider the two-agent push-box task again. The state space $\mathcal{S}$ is composed of the component set $\{V_1, V_2, V_3, V_4, V_5, V_6\}$, with three entity sets $\{V_1, V_2\}, \{V_3, V_4\}, \{V_5, V_6\}$ representing the location of agent one, agent two, and the box. Suppose the two agents are homogeneous. The component sets $\{V_1, V_2, V_5, V_6\}$ and $\{V_3, V_4, V_5, V_6\}$ are equivalent, because both of them consider the locations of one agent and the box jointly. Since the agents are homogeneous, it is irrelevant which agent is considered. Assigning different counters to equivalent subspaces will encourage an exploration policy to visit states that are visited by fellow homogeneous agents. This results in less efficient exploration. Therefore, equivalent subspaces share one common counter. The subspace $\mathcal{S}_k$ a counter $c_k$ is associated with is defined by $\mathcal{S}_k = \prod_{V_i \in D_{E_k}} V_i$, where $E_k$ is a component set of subspace $k$.

In addition, we also consider level 0, where the component set of each subspace has only one element. Empirically, we found that level 0 subspace leads to very efficient exploration in some tasks.

Note that this strategy of selecting subspaces is relatively simple still and does not scale well. We defer development of more complex selection strategies to future work. Here we are primarily interested in studying the efficacy of training an exploration strategy with such rewards, which we study in the next section.

## 4 EXPERIMENTAL RESULTS

We evaluate the proposed CMAE approach on two challenging environments: 1) the sparse-reward cooperative task from Wang et al. (2020); 2) the sparse-reward version of the Starcraft multi-agent challenge (SMAC) (Samvelyan et al., 2019). In both environments, agents need to coordinate their behavior over extended periods of time to obtain a reward.

**Environments:** We first consider the following four tasks on the sparse-reward environments provided by Wang et al. (2020):

- *Pass*: Two agents operate within two rooms of a $30 \times 30$ grid. There is one switch in each room, the rooms are separated by a door and agents start in the same room. The door will open only when one of the switches is occupied. The agents see collective positive reward and the episode terminates only when both agents changed to the other room. The task is considered solved if both agents are in the right room.
- *Secret-room*: Secret-room extends *Pass*. There are two agents and four rooms. One large room on the left and three small rooms on the right. There is one door between each small room and the large room. The switch in the large room controls all three doors. The switch in each small room only controls the rooms door. The agents need to navigate to one of the three small rooms, *i.e.* target room, to receive positive reward. The grid size is $25 \times 25$. The task is considered solved if both agents are in the target room.
- *Push-box*: There are two agents and one box in a $15 \times 15$ grid. Agents need to push the box to the wall to receive positive reward. The box is heavy, so both agents need to push the box in the same direction at the same time to move the box. The task is considered solved if the box is pushed to the wall.
- *Island*: Two agents, nine treasures, and a wolf operate in a $10 \times 10$ grid. Agents get a collective reward of 10 for collecting a treasure and a collective reward of 300 when crushing the wolf. The wolf and agents have maximum energy of eight and five respectively. The energy will decrease by one when being attacked. Therefore, one agent cannot crush the wolf. The agents need to collaborate to complete the task. The task is considered solved if the wolf is killed.

We also consider four tasks in SMAC (Samvelyan et al., 2019):

- *3m-dense*: There are three marines in each team. Agents need to collaboratively eliminate the three marines on the other team. An agent sees a reward of $+1$ when it causes damage

Figure 1: Results on *Pass*, *Secret room*, *Push-box*, and *Island* environment.

| Method (target success rate) | Ours | EITI | EDTI | IDQ | IDQ + RND |
|---|---|---|---|---|---|
| Pass (80%) | **2.61M±0.10M** | 384M±1.2M | 381M±2.8M | > 500M | > 500M |
| Secret-Room (80%) | **0.71M±0.05M** | 448M±10.0M | 382M±9.4M | > 500M | > 500M |
| Push-Box (10%) | **0.52M±0.04M** | 307M±2.3M | 160M±12.1M | > 500M | > 500M |
| Push-Box (80%) | **0.68M±0.02M** | 307M±3.9M | 160M±8.2M | > 500M | > 500M |
| Island (20%) | **7.50M±0.12M** | 480M±5.2M | 322M±1.4M | > 500M | > 500M |
| Island (50%) | **13.9M±0.21M** | > 500M | > 500M | > 500M | > 500M |

Table 1: Environment steps required to achieve the indicated target success rate on *Pass*, *Secret Room*, *Push-Box*, and *Island* environments.

to an enemy's health. A reward of $-1$ is received when its health decreases. All the rewards are collective. A reward of $+200$ is obtained when all enemies are eliminated.
- *8m-dense*: Similar to *3m-dense*, but with eight marines on each team.
- *3m-sparse*: Similar to *3m-dense*, but the reward is sparse. Agents only see reward $+1$ when all enemies are crushed.
- *8m-sparse*: Similar to *3m-sparse*, but with eight marines on each team.

**Experimental Setup:**

For the grid world task, we combine CMAE with Q-learning. For *Pass*, *Secret-room*, and *Push-box*, the Q value function is represented via a table. For *Island* we use a DQN (Mnih et al., 2013; 2015). The Q-function is parameterized by a three-layer perceptron (MLP) with 64 hidden units per layer and ReLU activation function. We compare CMAE with exploration via information-theoretic influence (EITI) and exploration via decision-theoretic influence (EDTI) (Wang et al., 2020), which are state-of-the-art algorithms on the four tasks. EITI and EDTI (Wang et al., 2020) results are obtained using the official code. For a more complete comparison, we also show the results of independent Q-learning (IDQ) with $\epsilon$-greedy and independent Q-learning with popular single agent exploration techniques, such as random network distillation (Burda et al., 2019).

For the SMAC tasks, we combine CMAE with the official code for QMIX (Rashid et al., 2018). We compare with MAVEN (Mahajan et al., 2019), QMIX (Rashid et al., 2018), QTRAN (Son et al., 2019), and VDN (Sunehag et al., 2018). All of the aforementioned methods reported impressive performance on the dense-reward version of SMAC. To our best knowledge, CMAE is the first to report results on the sparse-reward version of any SMAC tasks.

For all the experiments, we consider level 0 to level 3 subspaces. Please see the Appendix for more details.

**Evaluation Protocol:** To assess efficacy of CMAE we use the following evaluation procedure: we test the target policies in an independent test environment every 400k environment steps during the training. Each test consists of ten testing episodes. We repeat all experiments using three runs with different seeds.

**Results:** We first compare CMAE with EITI and EDTI on *Pass*, *Secret Room*, *Push-Box*, and *Island*. The results are summarized in Fig. 1, where test task success rate versus number of environment steps is shown. We observe CMAE to achieve a $100\%$ success rate on *Pass*, *Secret Room*, and *Push-Box* within 2M environment steps. In contrast, EITI and EDTI (Wang et al., 2020) need more than 300M steps to achieve an $80\%$ success rate (Tab. 1). In *Island*, CMAE achieves a success rate

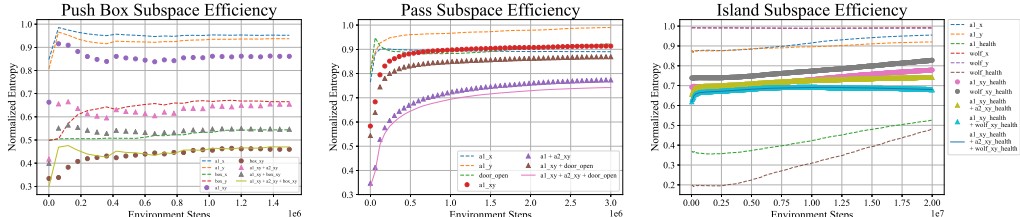

Figure 2: Results on SMAC: *3m-sparse*, *8m-sparse*, *3m-dense*, *8m-dense* environment.

Figure 3: Normalized entropy of different levels $l$ on *Push-box*, *Pass*, and *Island* environments.

(capture rate) above $50\%$ within 20M environment steps (Fig. 1). In contrast, EITI and EDTI need more than 480M and 322M steps to achieve a $20\%$ success rate (Tab. 1). The main reasons that EITI and EDTI need much more environment steps: they require a large number of samples to estimate the influence of one agent's behavior on other agents' between each update. Specifically, EITI and EDTI need 64,000 environment steps between each update, which makes them less sample efficient. IDQ with $\epsilon$-greedy and IDQ with RND does not achieve any success in those tasks.

On SMAC, we compare CMAE with MAVEN (Mahajan et al., 2019), QMIX (Rashid et al., 2018), QTRAN (Son et al., 2019), and VDN (Sunehag et al., 2018). The results are summarized in Fig. 2. In sparse-reward tasks, MAVEN, QMIX, QTRAN, and VDN have at most $2\%$ winning rate. In contrast, CMAE achieves a win rate higher than $80\%$. Recently, Taiga et al. (2020) point out that many existing exploration strategies excel in challenging sparse-reward tasks but fail in simple tasks that can be solved by using classical methods such as $\epsilon$-greedy. To ensure CMAE doesn't fail in simpler tasks, we run experiments on dense-reward SMAC tasks. As shown in Fig. 2, CMAE achieves similar performance to state-of-the-art baselines in simpler dense-reward SMAC tasks.

To investigate which subspaces CMAE selects, we plot the normalized entropy of different subspaces in Fig. 3. In the *Push-Box* task, CMAE mostly chooses the boxes location to explore. In the *Island* task, CMAE mostly explores the health of the wolf. For the *Pass* task, instead of exploring subspaces, CMAE explores the full global space.

We also compare the exploration behavior of CMAE to $\epsilon$-greedy using the *Secret room* environment. As shown in Fig. 4, early during training, both CMAE and $\epsilon$-greedy explore only locations in the left room. However, after $1.5$M steps, CMAE agents are able to frequently visit the right three rooms while $\epsilon$-greedy still mostly visits the left room.

Following the reviewers' suggestions, we also consider a shared count-based bonus on the group observations as a baseline. We study Q-learning with this shared count-based bonus on the group observations for the *Secret-room* and *Push-box* tasks. The shared count method achieves a $5.1\% \pm 1.3\%$ and $2.2\% \pm 1.1\%$ success rate on *Secret-room* and *Push-box* respectively. In contrast, our approach can achieve a $100\% \pm 0.1\%$ success rate in both tasks. The training curves are shown in Fig. 5. The count-based bonus method is sub-optimal because group observation is not necessarily the most efficient subspace to explore. This demonstrates the effectiveness of our subspace selection mechanism.

In addition, to demonstrate CMAE is applicable to a wide variety of challenging tasks. We conduct experiments on the SMAC *6h_vs_8z-dense (super hard)* and *6h_vs_8z-sparse (super hard)* tasks, where the opponent agents' AI are set to the 'super hard' level. In *6h_vs_8z-dense (super hard)*, an agent sees a reward of $+1$ when it causes damage to an enemy's health. A reward of $-1$ is received when its health decreases. In *6h_vs_8z-sparse (super hard)*, an agent can only see non-zero

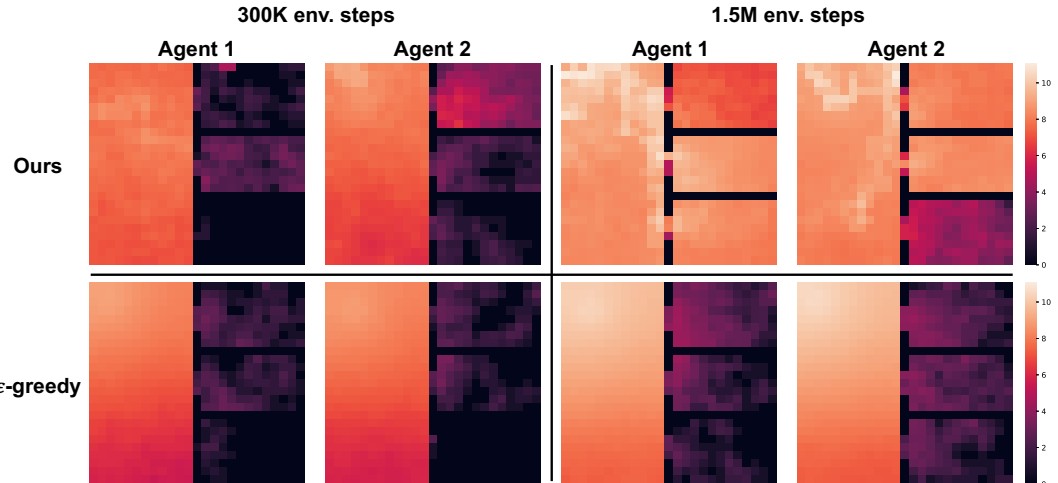

Figure 4: Visitation map of Ours (CMAE) and baseline on the *Secret-Room* environment.

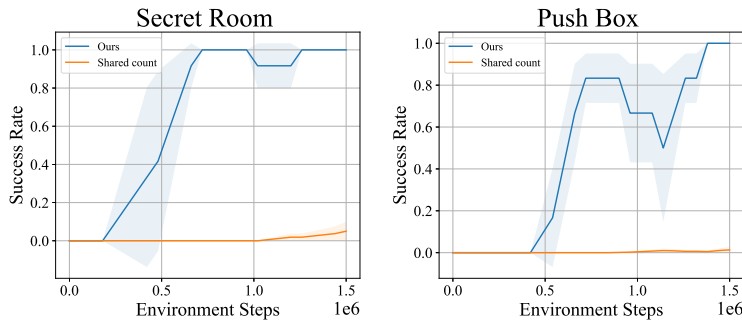

Figure 5: Results of our approach and shared count baseline on *Secret-room* and *Push-box*.

reward when an opponent is eliminated or a teammate is eliminated. We compare our approach to MAVEN Mahajan et al. (2019), which reports the state-of-the-art results on this task. All approaches are trained for 8M steps. In *6h_vs_8z-sparse (super hard)*, CMAE achieves a $45.6\% \pm 3.2\%$ win rate while MAVEN achieves a $4.3\% \pm 0.9\%$ win rate. In *6h_vs_8z-dense (super hard)*, CMAE and MAVEN achieve a $60.9\% \pm 1.3\%$ and $61.2\% \pm 2.3\%$ success rate respectively. This illustrates that dense reward environments tend to be easier than sparse ones. The training curves are shown in Fig. 6.

## 5 RELATED WORK

We discuss recently developed methods for exploration in reinforcement learning, multi-agent reinforcement learning and concurrent reinforcement learning subsequently.

**Exploration for Reinforcement Learning:** A wide variety of exploration techniques for deep reinforcement learning have been studied, deviating from classical noise-based methods. Generalization of count-based approaches, which give near-optimal results in tabular reinforcement learning, to environments with continuous state spaces have been proposed. For instance, Bellemare et al. (2016) propose a density model to measure the agent's uncertainty. Pseudo-counts are derived from the density model which give rise to an exploration bonus encouraging assessment of rarely visited states. Inspired by Bellemare et al. (2016), Ostrovski et al. (2017) discussed a neural density model, to estimate the pseudo count, and Tang et al. (2017) use a hash function to estimate the count.

Besides count-based approaches, meta-policy gradient Xu et al. (2018) uses the actor policy's improvement as the reward to train an exploration policy. The resulting exploration policy differs from the actor policy, and enables more global exploration. Stadie et al. (2016) propose an exploration strategy based on assigning an exploration bonus from a concurrently learned environment model. Lee et al. (2020) cast exploration as a state marginal matching (SMM) problem and aim to learn

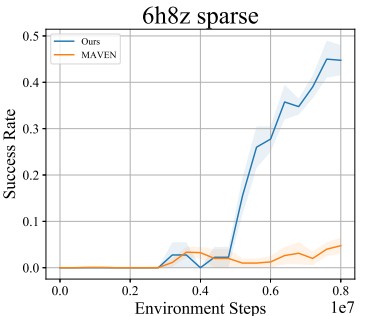 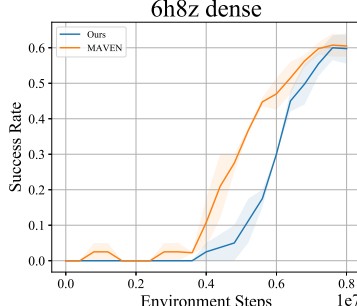

Figure 6: Results of our approach and MAVEN on *6h_vs_8z-sparse (super hard)* and *6h_vs_8z-dense (super hard)*.

a policy for which the state marginal distribution matches a uniform distribution. Other related works on exploration include curiosity-driven exploration Pathak et al. (2017), diversity-driven exploration Hong et al. (2018), GEP-PG Colas et al. (2018), $EX^2$ Fu et al. (2017), and bootstrap DQN Osband et al. (2016). In contrast to our approach, all the techniques mentioned above target single-agent deep reinforcement learning.

**Multi-agent Reinforcement Learning:** MADDPG Lowe et al. (2017) uses a central critic that considers other agents' action policies to handle the non-stationary environment issues in the multi-agent setting. DIAL Foerster et al. (2016) uses an end-to-end differentiable architecture that allows agents to learn to communicate. Jiang & Lu (2018) propose an attentional communication model that learns when communication is helpful for a cooperative setting. Foerster et al. (2017) add a 'fingerprint' to each transition tuple in the replay memory to track the age of the transition tuple and stabilize training. In 'Self-Other-Modeling' (SOM) Raileanu et al. (2018) an agent uses its own policy to predict others agents' behavior and states.

While inter-agent communication Lowe et al. (2017); Jiang & Lu (2018); Foerster et al. (2016); Rashid et al. (2018); Omidshafiei et al. (2017); Jain et al. (2019) has been considered, for exploration, multi-agent approaches rely on classical noise-based exploration. As discussed in Sec. 1, a noise-based approach prevents the agents from sharing their understanding of the environment. A team of cooperative agents with a noise-based exploration policy can only explore local regions that are close to their individual actor policy, which contrasts the approach from the proposed method.

Recently, approaches that consider coordinated exploration have been proposed. Multi-agent variational exploration (MAVEN) (Mahajan et al., 2019) introduces a latent space for hierarchical control. Agents condition their behavior on the latent variable to perform committed exploration. Influenced-based exploration (Wang et al., 2020) captures the influence of one agent's behavior on others. Agents are encouraged to visit 'interaction points' that will change other agents' behaviour.

**Concurrent Reinforcement Learning:** Dimakopoulou & Roy (2018) study coordinated exploration in concurrent reinforcement learning, maintaining an environment model and extending posterior sampling such that agents explore in a coordinated fashion. Parisotto et al. (2019) proposed concurrent meta reinforcement learning (CMRL) which permits a set of parallel agents to communicate with each other and find efficient exploration strategies. The concurrent setting fundamentally differs from the multi-agent setting of our approach. In a concurrent setting, agents operate in different instances of an environment, *i.e.*, one agent's action has no effect on the observation and rewards received by other agents. In contrast, in the multi-agent setting, agents use the same instance of an environment. An agent's action changes observations and rewards observed by other agents.

## 6 CONCLUSION

We propose coordinated multi-agent exploration (CMAE). It defines shared goals and learns coordinated exploration policies. We studied subspace selection which helps to find a goal for efficient exploration. Empirically, we demonstrate that CMAE increases exploration efficiency significantly. Compared to state-of-the-art baselines, CMAE needs only $1 - 5\%$ of the data to achieve similar or better results on various sparse-reward tasks. We hope this is a first step toward efficient coordinated MARL exploration. Going forward we will study more complicated subspace selection techniques and scale to more agents.

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
