# OpenReview forum: "Coordinated Multi-Agent Exploration Using Shared Goals"
_ICLR.cc/2021/Conference — Reject_

### Official Review · AnonReviewer2 · 2020-10-28
**Lack of clarity**

**Rating:** 4
**Confidence:** 4

**Review:**

## Overview

This paper proposes an exploration method for collaborative multi agent RL similar to count-based approaches in single-agent RL. It then provide some experimental analysis of the method in two kinds of environments (Starcraft and multi-agent particle-based environments)

Overall, the exposition of the method lacks clarity, and the experimental results suffer some limitations. Therefore, I am not recommending this paper for acceptation at ICLR 2021.

## Method

The algorithm considers all the variables of the problem $V_1,\cdots,V_M$, which can sometimes be grouped when they correspond to the same agents (eg there are two variables for the x,y coordinates of a given agent).
At the core, a counter is incremented when a given subset $\mathcal{S}_k$ ("a low dimensional subspace") of the variables reach a given "configuration", which is then used to identify subsets of variables that have rarely been attained. An intrinsic reward is then used to train a policy to reach these rare configurations.
Several things are unclear to me:
* The exploration are trained at the end of episode, and since the counts have changed since the last episode the target configuration (thus the reward) will change too. However it is not clear if the exploration policy is retrained from scratch or simply fine-tuned from the previous iteration
* It is not clear what a "configuration $s_k\in\mathcal{S_k}$" is. According to section 2.1, the variables $V_i$ are assumed to be in $\mathbb{R}$, thus finding the exact same configuration several times over the course of training seems unlikely. I assume there is some kind of discretization applied, but more details are required here. In particular, the exact way a configuraton is defined determines how many configurations are being tracked and thus the complexity of the algorithm (both in space and time), which is not discussed.


Section 3.2 describes how the subset of variables are being selected. Essentially, the sets are designed such that the variables of a given entity are always in the same subset, and then only subsets of entities of a limited size (l) are being considered.
In practice, the exact value of $l$ doesn't seem to be specified in the paper. Also lacking is the actual number of subspaces that the choice of $l$ incurs in each of the considered environments, and how the choice of $l$ influences the performance the performance of the algorithm (both in terms of final reward and runtime/memory)

Towards the end of section 3.2, one can read:

> In addition, we also consider level 0, where the component set of each subspace has only one element. Empirically, we found that level 0 subspace leads to very efficient exploration in some tasks

I don't quite understand what this level 0 corresponds to. Is it simply a subspace per variable (as opposed to a subspace per entity set, which in my understanding would be level 1) ?
As for the empirical claim, it doesn't seem to be backed by any experiments of the experimental section. We note that at one variable per variable, the algorithm would reduce to a standard count-based (single agent) exploration scheme. If such scheme are found to be efficient, they should be benchmarked in the experiments, which they currently aren't.


## Experiments

The authors provide experiments on two domains: some discrete multi-agent particle world environments and some starcraft scenarios (from SMAC). Overall, the experimental results suffer some limitations that undermine the strength of the claim.

### Particle environments

The environments are introduced in [1], and the authors directly compare to the results of the corresponding paper.
The proposed exploration method is applied on top of different base learning algorithms (sometimes vanilla tabular Q-learning, sometimes DQN). This inconsistency isn't discussed and make comparison between algorithms quite challenging since it adds a moving part.

The authors provide comparison with the methods proposed from [1], EDTI and EITI. However the comparison doesn't seem entirely fair. In [1], the authors reported (and presumably optimized for) the number of environment steps, while this papers reports the number of environment steps. These are two distinct metrics, and it is reasonable to believe that small adjustments would make EDTI and EITI much more competitive in terms of environment frames. For example, in [1] they used returns of length 2048 (at it is the default in PPO2), but based on ppo2 performance in environments like Atari, a much more conservative return length (say 128 or 256) could have sufficed, thereby drastically improving the sample complexity. Similarly, reducing the batch-size could improve the sample complexity as well. As it stands, the claim " Specifically, EITI and EDTI need 64,000 environment steps between each update, which makes them less sample efficient" isn't backed by any experimental evidence.
It is not clear how the number of environment-steps that are reported in table 1 are obtained. In particular, despite the fact that the experiments were run multiple times, the table doesn't reflect any kind of confidence interval. This is problematic because the environment "Island", for example, exhibits quite a high variance in Figure 1. While table 1 claims that 50% success rate is achieved at 13.9M steps, Figure1 shows that, on average, the success rates dips below 50% after 17M, which raises question on the stability of the learnt policy.

### Starcraft

The paper presents the results on the two easiest Starcraft environments of the SMAC set, either with sparse rewards (+1 for victory, -1 for defeat) or with dense rewards.
Contrary to the other environments, the authors don't provide a comparison with EDTI and EITI, which is unfortunate as they appear to be the strongest available baseline to date.
Amongst the reported baselines, CMAE is the only methods achieving decent win-rate on the sparse environments, and it appears to match the performance of the baselines on the dense counter-part, which is encouraging.
It has to be noted that, according to the training curves in appendix, it appears that the baselines learnt a fleeing strategy (ie never engaging the enemy, thus never "solving" the task), which, while clearly sub-optimal, is a local optimum. Perhaps a table reporting the average returns of each method could paint a more accurate picture.

Finally, it's unfortunate that the paper sticks to these two easy environments. Understanding the failure modes in the easiest environment that can't be solved by this method would yield significantly more insights on the methods than near-perfect scores on easy tasks. In particular, in the MAVEN paper [2], results on harder (dense) starcraft tasks were reported, and it would be valuable to see how the proposed method performs on those.


[1] Tonghan Wang, Jianhao Wang, Yi Wu, and Chongjie Zhang. Influence-based multi-agent explo-ration. InProc. ICLR, 2020
[2] Anuj Mahajan, Tabish Rashid, Mikayel Samvelyan, and Shimon Whiteson. MAVEN: multi-agentvariational exploration. InProc. NeurIPS, 2019

---

> ### Author Response · Authors · 2020-11-25
> **Response to AnonReviewer2**
>
> We thank the reviewer for the detailed comments. We think there are several misunderstandings that we point out below.
>
> Re 1: Value of $l$ (level).\
> For all the experiments, we consider level 0 to level 3 subspaces. We clarified in Sec 4.
>
> Re 2: No level 0 results?\
> Please see Fig. 3 (right), the level 0 subspace (wolf_health) in the *island* task has the lowest entropy throughout the training, and is selected as the subspace to explore. Thanks to the selected level 0 subspace, our approach is much more data efficient than baseline approaches on *island* (Tab. 1, Fig. 1).
>
> Re 3: Level 0 reduced to standard single-agent count-based exploration?\
> CMAE at level 0 differs from single-agent count-based exploration in two aspects. First, CMAE decouples the exploration and target policy, and uses the level 0 variable as a shared goal to guide coordinated exploration. In contrast, single-agent count-based exploration does neither consider any form of coordination between agents nor does it decouple exploration and target policies. Second, in level 0, we only consider one variable of the state space. In contrast, single-agent count-based exploration considers an agent’s local observation or the state, which contains more than one variable.
>
> Re 4: Usage of Q-learning and DQN.\
> We use Q-learning for *pass*, *secret room*, and *push box* simply because the number of states is small and tabular Q-learning is applicable. DQN is used for *island* because the number of states is larger. Therefore, we use a deep net to approximate the Q table. The fact that we use both demonstrates the generality of the proposed approach.
>
> Re 5: Comparison with [1].\
> [1] reports the number of model updates, which is not a standard metric for evaluating RL algorithms. The metric doesn’t reflect an RL approach’s data efficiency. In contrast, the number of environment steps is a standard metric which reflects the data efficiency and is adopted by most RL works [a, b, c, d, e], particularly works on RL exploration [f, g, h, i, j].
> Therefore, following most existing works, we report the number of environment steps, which we believe is the right metric for evaluating RL algorithms.
>
> Re 6: Small adjustments would make EDTI and EITI much more competitive in terms of environment frames. \
> Following the reviewer’s suggestion, we run the official code of EITI with a return length of 128 and 256 on *pass*. At 3M environment steps, EITI with 128 and 256 return length achieves 0%$\pm$ 0% success rate while our approach achieves  100% $\pm$ 0% success rate.
> After 590M environment steps of training, EITI with 128 and 256 return length achieves 0.0% $\pm$ 0% and 54.8% $\pm$ 12% success rate which is much lower than the success rate of 80% $\pm$ 2% achieved by using the default setting.
> This shows that decreasing the return length does not make EITI more competitive in terms of environment steps. Very much in contrast, the adjustment suggested by the reviewer undermines EITI’s performance.
>
> Re 7: Confidence interval of Table 1. \
> We added confidence intervals to Table 1.
>
> Re 8: Not comparing with EDTI and EITI on SMAC. \
> We compare our approach to the state-of-the-art methods, including MAVEN, QMIX and QTRAN, on SMAC. EDTI and EITI [1] only report results on 2D grid world tasks, and do not report results on any SMAC tasks. It is unclear why the reviewer claims that EDTI and EITI are the strongest baselines to date on SMAC. To the best of our knowledge there is no evidence for this claim.
>
> Re 9: Harder SMAC environment with dense reward. \
> We run our approach and MAVEN on *6h_vs_8z super hard* with dense reward. Following the MAVEN paper, both approaches are trained with 8M environment steps. Our approach and MAVEN achieves 60.9% $\pm$ 1.3% and 61.2% $\pm$ 2.3% success rate respectively. This demonstrates that in dense reward environments, our approach has similar performance to the state-of-the-art approach. The results are added to Sec. 4 and training curves are presented in Fig. 6.
>
> [a] Continuous control with deep reinforcement learning. Lillicrap et al. ICLR16\
> [b] Human-level control through deep reinforcement learning. Mnih et al. Nature17\
> [c] Scalable trust-region method for deep reinforcement learning using Kronecker-factored approximation, Wu et al. NeurIPS17\
> [d] Hindsight Experience Replay. Andrychowicz et al. NeurIPS17\
> [e] Model-based Policy Optimization with Unsupervised Model Adaptation. Shen et al. NeurIPS20\
> [f] MAVEN: Multi-Agent Variational Exploration. Mahajan et al. NeurIPS19\
> [g] Optimistic Exploration even with a Pessimistic Initialisation. Rashid et al. ICLR20\
> [h] On Bonus Based Exploration Methods In The Arcade Learning Environment. Taiga et al. ICLR20\
> [i] Curiosity-driven Exploration by Self-supervised Prediction. Pathak et al. ICML17\
> [j] #Exploration: A Study of Count-Based Exploration for Deep Reinforcement Learning. Tang et al. NeurIPS17

---

### Official Review · AnonReviewer4 · 2020-10-28
**Unconvinced that single-agent exploration bonuses applied to the group aren't enough in the CTDE setting**

**Rating:** 6
**Confidence:** 4

**Review:**

Summary of paper:
The paper introduces an exploration bonus tailored to multi-agent learning in the CTDE (centralized training, decentralizing execution) setting. The bonus works by: 1) dividing up the observation space into subspaces (in this case, corresponding to the entities, pairs of entities, triplets of entities, etc), 2) maintaining running counters within each subspace of every possible configuration within that subspace, 3) identifying the lowest entropy subspace, 4) sampling the replay buffer to find a rarely occurring (but importantly, reachable) “goal” state within the lowest entropy subspace, and finally, 5) rewarding the exploration policy for visiting the goal state. The authors test their method in two domains: 1) a series of coordinated multi-agent exploration challenges in grid worlds, and 2) the Starcraft Multi-Agent Challenge (SMAC). They demonstrate much faster learning over a series of baselines.

Pros:
The paper is generally well-written. The paper uses interesting tasks, and the experiments are generally well-structured. The experiments include modern baselines, and the sample efficiency over the baselines is impressive (at least at first glance).

Cons:
1) I don’t think the authors adequately convinced me that there is a new problem to solve here that couldn’t be tackled with straightforward applications of single-agent exploration methods in the multi-agent setting. I assume their IDQ + RND baseline applies the RND bonus independently to each agent (i.e. each agent has a separate target network to predict)? That is a useful baseline for sure, but its not exactly very strong. Why not use a single RND bonus for the *group* observations to induce correlated exploration? Since this is the CTDE setting, single-agent exploration bonuses can be applied somewhat straightforwardly to the group. Even simpler, why not include a simple count-based bonus on the group observations (i.e. r(s)=1/sqrt(c(s)), where c(s) is the count of the group observation s)? The authors’ method is more or less a group count-based exploration bonus that also takes advantage of structured observations, so this baseline is both possible and relevant.
2) The authors’ method seems to me to build in a lot more hand-crafted knowledge than they let on (and more than any of their baselines, as far as I can tell). Their method requires structured knowledge of the agent’s observations (e.g. these variables correspond to these entities), which is not always available (e.g. from pixels). Can the authors comment on the fairness of comparing their method to the baselines they use, as well as the scalability to less structured observation spaces?
3) Additionally, it looks to me like their method would be fooled by a large number of unreachable states for one entity - in this case, Hmax would be large, so the normalized entropy would be small, and the agent would keep exploring a particular entity set of variables, without progress. For example, imagine a world with two boxes, one which is movable, and one which is not, but imagine the agent does not know in advance that one is unmovable. CMAE would focus on the unmovable box forever (granting a fixed useless exploration bonus for the single state the box has ever and will ever occupy). Can the authors comment on this?

Questions:
1) I understand how counting observations works in the grid worlds, but isn’t SMAC a continuous state space? How does counting work there? More generally, can you comment on how the method would work in continuous state spaces (e.g. binning, neural density models, etc)?
2) Maybe I missed it, but how many subspaces (K) are there in these experiments? The authors state that 2^M is “clearly intractable” (where M is the number of component subspaces) but seem to imply that 2^N is tractable (where N is the number of entity subspaces). However, M and N are not obviously very different numbers, depending on the number of components per entity.
3) Deterministically focussing exploration on the lowest entropy subspace seems potentially unstable (e.g. in presence of many unreachable states - see comment above). Did the authors consider probabilistic subspace selection, in which the subspace is chosen from, say, a softmax distribution over the subspace entropies?
4) In the discussion of “Exploration for Reinforcement Learning”, the authors state “In contrast to our approach, all the techniques mentioned above target single-agent deep reinforcement learning.” Strictly speaking, this is true, but can’t most (if not all) single-agent exploration techniques be applied to multi-agent in the CTDE setting by treating the group as the agent?

Typos/formatting:
1) Figure 3 legends are nearly impossible to read. Find space to increase their size.
2) Figure 2 legends are a) too small and b) the same across all 4 subplots, so just use one large legend on the right.
3) In paragraph under Figure 3: “SMAC achieves similar performance…” -> “CMAE achieves similar performance…”
4) In the Multi-Agent RL part of Related Work, there are some citep<->citet mishaps.
5) Conclusion: “increase” -> “increases”

---

> ### Author Response · Authors · 2020-11-25
> **Response to AnonReviewer4**
>
> We thank the reviewer for the detailed comments.
>
> Re 1: Why not include a single-agent count-based bonus on the group observations. \
> Following the reviewer’s suggestion, we run Q-learning with a shared count-based bonus on the group observations for the *secret room* and *push box* tasks. The shared count method achieves 5.1% $\pm$ 1.3% and 2.2% $\pm$ 1.1% success rate on *secret room* and *push box* respectively. In contrast, our approach can achieve a 100% $\pm$ 0.1% success rate in both tasks. The count-based bonus method is suboptimal because group observation is not necessarily the most efficient subspace to explore. This demonstrates the effectiveness of our subspace selection mechanism. The results are added to Sec. 4 and training curves are shown in Fig. 5.
>
> Re 2: Use a lot of domain knowledge of handcrafted features. \
> The only information we need is the component set of the state space. This information is provided by most of the existing multi-agent RL environments, such as the StarCraft multi-agent challenge (SMAC) [a], the multiple-particle environment (MPE) [b], the Multiagent emergence environments [c], Google Research Football (GFootball) [d], and MuJoCo Soccer [e]. The proposed approach leverages the component set information to build a useful inductive bias into the algorithm and thus achieves more efficient exploration. In contrast, prior MARL approaches for exploration don’t leverage the valuable information provided by the environment.
>
> Re 3: CMAE would be fooled by a large number of unreachable states for one entity. \
> Thanks for pointing this out. Indeed, if a variable remains constant in the state space, CMAE may stick with this variable. However, adding a constant variable in a state representation doesn’t seem to be a reasonable MARL environment design. Empirically, we don’t observe any of these constant variables in multi-agent RL environments such as SMAC [a] and MPE [b].
>
> Re 4: How would the method work in continuous state spaces. \
> In continuous spaces, counting could be performed by either binning or any off-the-shelf continuous counting methods such as work by Tang et al. [f] or the neural density model [g]. Empirically, we found our approach with binning achieves good results on SMAC 3m-sparse and 8m sparse tasks, where the state space is continuous. Please see Fig. 2 of the paper for more details.
>
> Re 5: How many subspaces (K) are there in these experiments. \
> The subspaces and their normalized entropies on *push box*, *pass*, and *island* are presented in Fig. 3. We consider level 0 to level 3 subspaces, which results in 9, 7, and 11 subspaces on *push box*, *pass*, and *island* tasks.
> For SMAC 3m and 8m tasks, we consider level 0 to level 3 subspaces, which results in 11 subspaces in both tasks.
>
> Re 6: Probabilistic subspace selection. \
> Thanks for the suggestion. We think probabilistic subspace selection is a good strategy when faced with environments with constant variables. We added a discussion on probabilistic subspace selection in Sec. 3.1.
>
> Re 7: Can’t most single-agent exploration techniques be applied to multi-agent in the CTDE setting by treating the group as the agent? \
> Please see Re 1.
>
> [a] The StarCraft Multi-Agent Challenge. Samvelyan et al. arxiv.19\
> [b] Multi-Agent Actor-Critic for Mixed Cooperative-Competitive Environments. Lowe et al. NeurIPS17\
> [c] Emergent Tool Use From Multi-Agent Autocurricula. Baker et al. ICLR20\
> [d] Google Research Football: A Novel Reinforcement Learning Environment. Kurach et al. AAAI20\
> [e] Emergent Coordination through Competition. Liu et al. ICLR19\
> [f] #Exploration: A Study of Count-Based Exploration for Deep Reinforcement Learning. Tang et al. NeurIPS17\
> [g] Count-Based Exploration with Neural Density Models. Ostrovski et al. ICML17

---

### Official Review · AnonReviewer3 · 2020-10-29
**Proposed technique has limited applicability**

**Rating:** 5
**Confidence:** 4

**Review:**

The paper proposes to improve the exponential sample complexity of finding a coordinated multi-agent strategy by learning an exploration policy for each agent that conditions on a shared goal. The exploration policy is mixed with the normal RL policy according to a parameter alpha, which is scaled down over time. The shared goal that agents pursue is selected by using an explicit counter mechanism over objects in the environment.

Strengths:
- The paper is well written.
- The reduction in sample complexity due to this technique is very large.
- Algorithms 1 and 2 are clear.

Weaknesses:
- The proposed counter mechanism relies on being able to manually identify entities in the environment, such as the box in the push-box environment. This has limited applicability to real-world problems with large-dimensional or visual state spaces, in which entities are not obvious a priori. Being able to explicitly count the number of times an agent has experienced an entity in a specific configuration is not a realistic expectation for interesting, real-world problems. Therefore, it is unclear how this method can be applied beyond simple tabular settings and video games.
- Similarly, it seems that deciding which subspaces are equivalent requires a significant amount of domain knowledge into each problem, and does not seem to be generally applicable.
- Why not benchmark against QMIX + RND, since both are tested independently?

Other suggestions:
- Typo on p. 3 "tuple is then store in a replay memory" -> stored
- Why was the number of steps between updates for EITI and EDTI held constant at 64,000? How many steps between updates were used for the proposed technique?

---

> ### Author Response · Authors · 2020-11-25
> **Response to AnonReviewer3**
>
> We thank the reviewer for the detailed comments.
>
> Re 1: Identify entity (component set) in the state space.\
> Information about component sets of a state space is provided by most of the existing multi-agent RL environments, such as the StarCraft multi-agent challenge (SMAC) [a], the multiple-particle environment (MPE) [b], the multi-agent emergence environments [c], Google Research football (GFootball) [d], and DeepMind MuJoCo soccer [e]. We think the proposed technique is hence widely applicable.
> CMAE leverages the component set information to build useful inductive biases into our algorithm and thus achieves more efficient exploration. In contrast, the existing MARL approaches for exploration don’t use this valuable information provided by the environment.
>
> Re 2: Cannot scale to visual state space. \
> Scaling MARL approaches to pure visual state spaces is an open problem. To our best knowledge, the state-of-the-art MARL approaches, such as Weighted QMIX [f], MAVEN [g], QMIX [h], MADDPG [b], do not provide any experimental results on environments with pure visual state spaces.
>
> Re 3: Counting in continuous state space.\
> In continuous spaces, counting could be performed by either binning or any off-the-shelf continuous counting methods such as the one proposed by Tang et al. [i] or the neural density model [j]. Empirically, we found our approach with binning achieves good results on SMAC 3m-sparse and 8m-sparse, where the state space is continuous. Please see Fig. 2 of the paper for more details.
>
> Re 4: Selecting subspaces requires domain knowledge.  \
> As discussed in Re 1, the only information we need is the component set of the state space. This information is provided by most of the MARL environments.
>
> Re 5: Benchmark against QMIX + RND. \
> We run QMIX with RND on the SMAC 8m-sparse task. Both our approach and QMIX + RND are trained for 2M environment steps. Our approach achieves 80.1% $\pm$ 1.3% win rate while QMIX + RND achieves only 1.5% $\pm$ 0.4% win rate. We added the results to Sec. B of the appendix.
>
>
> [a] The StarCraft Multi-Agent Challenge. Samvelyan et al. arxiv.19\
> [b] Multi-Agent Actor-Critic for Mixed Cooperative-Competitive Environments. Lowe et al. NeurIPS17\
> [c] Emergent Tool Use From Multi-Agent Autocurricula. Baker et al. ICLR20\
> [d] Google Research Football: A Novel Reinforcement Learning Environment. Kurach et al. AAAI20\
> [e] Emergent Coordination through Competition. Liu et al. ICLR19\
> [f] Weighted QMIX: Expanding Monotonic Value Function Factorisation for Deep Multi-Agent Reinforcement Learning. Rashid et al. NeurIPS20\
> [g] MAVEN: Multi-Agent Variational Exploration. Mahajan et al. NeurIPS19\
> [h] QMIX: Monotonic Value Function Factorisation for Deep Multi-Agent Reinforcement Learning. Rashid et al. ICML18\
> [i] #Exploration: A Study of Count-Based Exploration for Deep Reinforcement Learning. Tang et al. NeurIPS17\
> [j] Count-Based Exploration with Neural Density Models. Ostrovski et al. ICML17

---

> > ### Comment · AnonReviewer3 · 2020-11-25
> > **Thank you for the response.**
> >
> > Thank you for the response, and for providing the results of the QMIX+RND comparison. I think that strengthens the paper.
> >
> > >Information about component sets of a state space is provided by most of the existing multi-agent RL environments, such as the StarCraft multi-agent challenge (SMAC) [a], the multiple-particle environment (MPE) [b], the multi-agent emergence environments [c], Google Research football (GFootball) [d], and DeepMind MuJoCo soccer [e]. We think the proposed technique is hence widely applicable.
> >
> > Although these environments are used as test beds to experiment with multi-agent algorithms, the eventual goal of multi-agent RL research should be to provide value on real world problems. The issue with the algorithm proposed in this paper is that it depends on an environment which provides component sets, which is an unrealistic expectation, and prevents it from scaling to the real world.
> >
> > > Scaling MARL approaches to pure visual state spaces is an open problem. To our best knowledge, the state-of-the-art MARL approaches, such as Weighted QMIX [f], MAVEN [g], QMIX [h], MADDPG [b], do not provide any experimental results on environments with pure visual state spaces.
> >
> > This is a good point. However, there is no reason, in principle, why algorithms like QMIX or MADDPG cannot be applied to problems with a high dimensional state space. Whereas the proposed algorithm suffers from this limitation.
> >
> > > In continuous spaces, counting could be performed by either binning or any off-the-shelf continuous counting methods such as the one proposed by Tang et al. [i] or the neural density model [j]. Empirically, we found our approach with binning achieves good results on SMAC 3m-sparse and 8m-sparse, where the state space is continuous. Please see Fig. 2 of the paper for more details.
> >
> > I have made note of that experiment. One suggestion for future work would be to combine your approach with something like a neural density model. This could help show how it can be extended beyond simulated games.

---

> > > ### Author Response · Authors · 2020-11-25
> > > **Response to AnonReviewer3**
> > >
> > > We thank the reviewer for the comments.
> > >
> > >
> > > >Although these environments are used as test beds to experiment with multi-agent algorithms, the eventual goal of multi-agent RL research should be to provide value on real world problems. The issue with the algorithm proposed in this paper is that it depends on an environment which provides component sets, which is an unrealistic expectation, and prevents it from scaling to the real world.
> > >
> > >
> > > Even in a real-world setting component sets can be available. For instance, in autonomous driving we may use object detectors to identify other agents and participants that are part of an environment. We remain convinced that this is a reasonable assumption to get research started. Should it be generalized? Absolutely. But do we have to solve all problems at once? We think the answer is no. The paper shows that an available decomposition can significantly improve algorithmic effectiveness. If the community is as excited about this result as we are, together, we can leverage component sets even in real-world settings.
> > >
> > >
> > > >This is a good point. However, there is no reason, in principle, why algorithms like QMIX or MADDPG cannot be applied to problems with a high dimensional state space. Whereas the proposed algorithm suffers from this limitation.
> > >
> > >
> > > The algorithm may not be directly scalable to a pure visual state-space but vanilla QMIX and MADDPG cannot be effectively applied in this setting either. Scalability challenges are due to different reasons, but this shouldn’t, according to our opinion, preclude a discussion about algorithms, their benefits and disadvantages.
> > >
> > >
> > > >I have made note of that experiment. One suggestion for future work would be to combine your approach with something like a neural density model. This could help show how it can be extended beyond simulated games.
> > >
> > >
> > > Thanks for sharing this thought which we agree is an interesting direction for future work. We believe that the careful analysis of the approach without this component (as presented in the current draft) still remains useful as it demonstrates why your thought of a neural density model should work.

---

### Official Review · AnonReviewer1 · 2020-10-30
**The proposed method outperforms prior work, but does not seem scalable to larger environments**

**Rating:** 5
**Confidence:** 3

**Review:**

---Post rebuttal---

Thank you for the detailed response. My main concern was regarding the scalability of the method to larger environments, e.g. w/ visual state space. I agree with the other reviewers regarding limited applicability of the method, and maintain my original score (Weak Reject).

---

The paper introduces a coordinated multi-agent exploration method that selects goals based on normalized entropy from multiple projected state spaces. The method greatly outperforms other algorithms on sparse-reward environments.

Weaknesses:

1. The process of selecting subspaces in the CMAE algorithm (Section 3.2) is not scalable to larger environments. I’d be willing to raise my score if, in more complex environments, using level 0 subspace still results in improved performance over existing work.

2. The related works section seems incomplete. It discusses single-agent exploration, concurrent RL, and multi-agent noise-based exploration (p.8), but these categories do not cover the following papers:
[1] Concurrent Meta Reinforcement Learning (Parisotto et al., 2019): Parallel multi-agent exploration in the same environment using shared memory and communication.
[2] Efficient Exploration via State Marginal Matching (Lee et al., 2019): Multi-task/multi-agent exploration that learns multiple “skills” (or “agents”) in the same environment, that together match some target state density.

Minor typo:
- “The resulting experience tuple is then stored* in a replay memory” (p.3)

---

> ### Author Response · Authors · 2020-11-25
> **Response to AnonReviewer1**
>
> We thank the reviewer for the detailed comments.
>
> Re 1: More complex environment.\
> We conduct experiments on a sparse reward version of SMAC 6h_vs_8z (super hard) task. Agents can only see non-zero reward when an opponent is eliminated or a teammate is eliminated. We compare our approach to MAVEN. All approaches are trained for 8M steps. Our approach with level 0 exploration achieves 45.6% $\pm$3.2%  win rate while MAVEN achieves 4.3%$\pm$0.9% win rate. The results are added to Sec. 4 and training curves are shown in Fig. 6.
>
> Re 2: Related work section is incomplete. \
> We added discussions about the papers suggested by the reviewer. See Sec. 5 for details.

---

### Decision · Program_Chairs · 2021-01-07
**Final Decision**

**Decision:**

Reject

**Comment:**

The paper presents an approach to multi-agent coordination using goal-driven exploration on subspaces of the observation space.

The results of the paper show that the authors' approach performs baselines on grid worlds and two tasks from the StartCraft Multi-agent Challenge. While the rebuttal clarified many points raised by the reviewers, there was an agreement that the paper should be more convincing regarding the applicability of the approach. The reviewers were concerned with the scalability of the approach to larger environment, as well as the amount of hand-crafting/domain knowledge required to apply the approach. Overall, while the paper contributes interesting results showing that such domain knowledge can help when properly leveraged, it feels like the approach needs be validated on more challenging environments before acceptance.